

# Machine learning-assisted genomic profiling to identify differences between Bacillus Calmette-Guérin (BCG) vaccine strains and non-BCG wild-type *Mycobacterium bovis*

Yunyun Shi[1,2,*], Jiang Yuan[3], Xiaobin Tang[3], Hai Luo[1], Genyun Tang[1] and Zhiyong Shen[1,2,*]

[1] School of Basic Medical Sciences, Huaihua Key Laboratory of Ion Channels and Complex Diseases, Hunan University of Medicine, Huaihua, Hunan Province, China
[2] Shenzhen Institute of Advanced Technology, Chinese Academy of Sciences, Shenzhen, Guangdong Province, China
[3] Nanshan District Maternal and Child Health Hospital, Shenzhen, Guangdong Province, China
* These authors contributed equally to this work.

## ABSTRACT

**Background:** Distinguishing Bacillus Calmette-Guérin (BCG) vaccines from pathogenic *Mycobacterium bovis* is critical in neonatal diagnostics, particularly where polymerase chain reaction (PCR) methods fail to detect key genomic variations in tuberculosis (TB)-endemic regions.

**Methods:** We developed a machine learning framework analyzing 72 clinical isolates (28 BCG, 44 non-BCG) using whole-genome sequencing. Two classifiers were implemented: a random forest optimized by out-of-bag error minimization and a one-dimensional convolutional neural network (1D CNN) with dropout regularization (0.3–0.5). Feature selection through permutation testing and gradient activation mapping enhanced interpretability.

**Results:** Cross-validation demonstrated robust performance for both models: the random forest achieved 96% accuracy using 47 BCG attenuation-related genes, while the convolutional neural network (CNN) maintained high generalizability with 95.8% (±3.4%) mean accuracy and perfect recall across stratified five-fold validation, supported by strong discriminative capacity (mean area under the curve (AUC): 0.964 ± 0.046). Key biomarkers included metabolic reprogramming (ko01100) and secondary metabolite biosynthesis (ko01110) pathways.

**Conclusion:** This genomic approach resolves BCG diagnostic ambiguities through conserved attenuation markers, with the 47-gene panel enabling rapid assays that reduce neonatal overtreatment risks. Future integration of transcriptomic data could optimize these biomarkers for clinical deployment in high-TB-burden settings.

Corresponding author
Zhiyong Shen,
szypanther@gmail.com

## INTRODUCTION

The Bacillus Calmette-Guérin (BCG) vaccine, created from a weakened form of *Mycobacterium bovis* bacteria, continues to serve as essential protection against tuberculosid (TB) in newborns, particularly where the disease remains widespread. Given soon after delivery, this vaccine effectively lowers infants' risk of developing severe systemic TB or meningitis (*World Health Organization (WHO), 2023*). However, its routine use creates diagnostic complexities when vaccinated infants later exhibit symptoms resembling TB or return positive screening results. Common immunological tests like tuberculin skin checks and interferon-gamma blood assays frequently show false positives due to the vaccine's induced immune responses (*Pai et al., 2014*). Even advanced molecular methods such as polymerese chain reaction (PCR) and GeneXpert MTB/RIF face challenges in separating genetic material from the vaccine strain *vs.* actual *M. bovis* or tuberculosis bacteria (*Nicol et al., 2011*). Such diagnostic uncertainties carry serious consequences that delayed treatment for genuine infections or unnecessary anti-TB medications, both posing substantial risks to fragile infant health.

Genetic differences between BCG vaccine strains and natural *M. bovis* could help address this diagnostic puzzle. Through historical laboratory modifications, BCG lost key genetic components like the RD1 virulence region critical for disease-causing ability in wild bacteria (*Brosch et al., 2007*). While these changes should theoretically allow strain differentiation, standard diagnostics focus too narrowly on individual markers like RD1 deletion. This limited approach ignores the diversity among BCG subtypes used globally. For instance, certain BCG subvariants missing the RD16 region may trick tests designed for other genetic targets (*Behr et al., 1999*). Additionally, unexpected genetic exchanges between bacterial strains or parallel evolution in clinical *M. bovis* isolates can blur the boundaries between vaccine and pathogen, reducing the reliability of standard PCR methods (*Smith et al., 2005*).

New approaches combining comprehensive genetic analysis with computational pattern detection may overcome these hurdles. Full genetic sequencing technologies now allow detailed comparison of entire bacterial genomes, while machine learning algorithms can detect subtle variations invisible to conventional methods. Similar computational tools have successfully classified tuberculosis strains and predicted antibiotic resistance by analyzing combinations of genetic mutations (*Coll et al., 2014*). Applying these techniques to BCG and wild *M. bovis* could reveal unique biomarkers-perhaps in non-coding DNA regions or interacting gene networks-that distinguish vaccine strains from dangerous pathogens. Such biomarkers might include non-coding regulatory elements, repetitive sequences, or epistatic interactions that are invisible to targeted assays (*Dheda et al., 2017*).

Our team is working to develop a genetic analysis tool that can reliably tell apart BCG vaccine reactions from actual *M. bovis* infections in newborns. The approach involves digging into full genetic blueprints from various BCG strains and comparing them with samples from confirmed infections. What we're looking for are consistent genetic markers that only show up in vaccine strains—think of them as microscopic fingerprints left by the weakened bacteria used in BCG vaccination. This isn't just about solving a tricky

diagnostic problem. It's showing how computer-powered biology can turn complex genetic data into real-world medical decisions. If successful, doctors in TB-prone regions could finally cut through the confusion between normal vaccine responses and dangerous infections. That means fewer babies getting unnecessary drugs they don't need, and faster treatment for those who truly require it-a critical improvement for overburdened healthcare systems.

## MATERIALS AND METHODS

### Strain selection and genome data

In this study, we systematically searched the NCBI database using the keyword "*Mycobacterium tuberculosis variant bovis*" and identified 76 high-quality genome assemblies at the scaffold level or higher. Among these, 28 genomes were explicitly designated as BCG derivatives in their nomenclature and were included in subsequent analyses. Additionally, four genomes lacking BCG designation in their names but annotated with BCG-related information in the "Modifier" field were classified as taxonomically ambiguous samples. Furthermore, 44 confirmed non-BCG wild-type mutant strains were identified for comparative analysis (Fig. 1). In the end, the accession numbers and information for the 72 clearly taxonomically classified genomes used can be found in Excel File S1).

### Computational infrastructure

All computational analyses were executed on a high-performance computing cluster utilizing two-socket Intel Xeon Gold 6230 processors (2.1 GHz base/3.9 GHz Turbo) with 40 physical cores (80 hyperthreaded logical processors) per node, Non-Uniform Memory Access (NUMA)-optimized memory allocation across two domains, and 1 TB of 2933 MHz DDR4 ECC RAM in 6-channel configuration. Workflow orchestration was implemented through Snakemake v7.32 with Docker 24.0 containers (CentOS 8.3 base), where the TensorFlow 2.15.0 framework exploited CPU-level parallelism *via* Intel MKL-DNN acceleration.

### Data preprocessing

The preprocessing pipeline focused on high-quality genome assemblies (≤200 contigs/genome) directly retrieved from NCBI, beginning with assembly integrity validation through (1) cross-referencing NCBI metadata for completeness flags and contiguity metrics (N50 ≥50 kb), (2) taxonomic confirmation *via* genome-wide average nucleotide identity (ANI ≥98% against *M. bovis* reference genomes). Raw read processing was omitted as analysis utilized pre-assembled genomes bearing NCBI's International Nucleotide Sequence Database Collaboration (INSDC) quality certification (contamination scores <1%, assembly gaps <5%).

### Bitscore values calculation of core orthologous genes

We annotated the 72 selected genome sequences using Prokka (*Seemann, 2014*), generating structural annotation files in General Feature Format (GFF) format and protein sequence files in FAA format. Subsequently, a pan-genome comparative analysis was

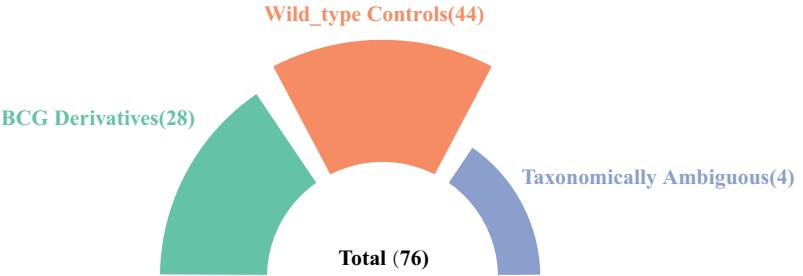

**Figure 1  A summary of the strain genomes selected in this study.**

performed with Roary (*Page et al., 2015*) (default parameters) based on the 72 GFF files, yielding a corresponding gene presence/absence matrix (gene_presence_absence.csv).

For further investigation, the DeltaBS approach was employed (*Gardner BinfLab, 2021*). This method utilizes profile hidden Markov models (HMMs) to integrate sequence diversity of homologous genes across organisms and capture natural variation patterns. Specifically, the profile HMMs of Gammaproteobacterial proteins (gproNOG.hmm) were retrieved from the eggNOG database (http://eggnog5.embl.de/) as a reference set of conserved bacterial orthologous groups (OGs). Although taxonomically labeled for Gammaproteobacteria, these OGs represent a subset of bacterial OGs (bacNOG) conserved across phyla including Actinobacteria. Annotated protein sequences from Prokka were aligned to their corresponding HMM profiles using the hmmsearch tool in the HMMER3.0 package (http://hmmer.org), generating bitscore values for each sequence. The DeltaBS method does not require phylum-matched HMMs as it uses these profiles as universal references to measure gene conservation. Core orthologous genes of each strain were then extracted using the parse_hmmsearch.pl and parse_bitscores.pl scripts from DeltaBS (*Gardner BinfLab, 2021*) to construct the DBS metric table (bitscores.tsv). This table served as the input for training and constructing a random forest classifier.

## Random forest classifier constructing and training

Given that the 72 genomes collected in this study represent unambiguously classified BCG vaccine strains and nonBCG wild-type *Mycobacterium bovis*, we use all the 28 complete BCG representative genomes and 44 non-BCG wild-type mutant strains genomes to develop a random forest classifier model for distinguishing between the two groups. Training the model on a set of 3,595 orthologous genes enabled effective differentiation between BCG vaccine strains and non-BCG wild-type *Mycobacterium bovis*. Model performance was evaluated using out-of-bag (OOB) accuracy as the primary metric. The random forest classifier was implemented using the randomForest and caret packages in R. To optimize model parameters, we systematically evaluated tree configurations. The number of trees (ntree) was set to 10,000, as the out-of-bag (OOB) error rate stabilized beyond this threshold. For mtry (the number of genes randomly sampled as candidate predictors at each node), a grid search was performed across values {1, n/10, n/5, n/3, n/2, n}, where n represents the total number of predictor genes (3,595). mtry = n/10 was selected to minimize collinearity among predictors, yielding the lowest OOB error (0.2).

Model refinement was achieved through iterative feature selection:

(1) Initial model (Model 1): All 3,595 genes were included.
(2) Sparsity pruning: Predictors with variable importance (VI) ≤ 0 were iteratively removed.
(3) Iterative optimization: The model was retrained using the reduced gene set, followed by exclusion of the lowest 50% VI-ranked genes.

This process was repeated until perfect OOB accuracy was attained.

To assess the null hypothesis (no association between predictors and strain type), 1,000 permuted datasets were generated by randomizing labels. The empirical *p*-value was calculated as the proportion of permuted models achieving accuracy equal to or greater than the original model. Top discriminative genes from the final model were functionally categorized *via* COG and KEGG databases using COG (https://github.com/transcript/COG) (*Westreich, 2019*) and KAAS (KEGG Automatic Annotation Server https://www.genome.jp/tools/kaas/) separately (*Moriya et al., 2007*).

## Convolutional neural network based genomic classifier construction based on top predicted genes

The convolutional neural network (CNN)-based genomic classifier was implemented using TensorFlow/Keras for binary classification of BCG strains. Input genomic features (47 numerical columns, 1D vectors) underwent preprocessing: missing values were replaced by feature-wise medians, followed by standardization *via* StandardScaler. The architecture comprised two cascaded convolutional blocks—each with Conv1D layers (32 and 64 filters, kernel size = 3, ReLU activation), MaxPooling1D (pool size = 2), and dropout regularization (rates = 0.3 and 0.5)—followed by a flattening layer and two dense layers (32 ReLU units and 1 sigmoid unit). Training utilized the Adam optimizer (learning rate = 0.001) with binary cross-entropy loss over 30 epochs (batch size = 32), while addressing class imbalance through inverse-frequency class weights computed by compute_class_weight(). Model robustness was rigorously evaluated through stratified five-fold cross-validation to preserve label distribution across all partitions. Performance metrics (accuracy, precision, recall, F1, area under curve-receiver operating characteristic (AUC-ROC)) and composite scores (weighted averages with default equal weights) were quantified across all validation folds, with final results reported as mean ± standard deviation. The AUC-ROC analysis was performed using roc_curve() from scikit-learn, with mean ROC curves and standard deviation bands generated through interpolation at 100 evenly spaced false positive rate thresholds. Following comprehensive cross-validation, the final model was trained on the complete dataset and serialized as an H5 file for deployment.

## Selection and evaluation method

The random forest model was selected for its transparent decision-making process, capacity to manage complex genomic datasets, and resilience to correlated genetic features, while its gene-ranking capability facilitated biological interpretation. The CNN framework

was implemented to identify nucleotide-level signatures within linear genomic data, utilizing layered filters to detect localized sequence motifs. To mitigate model overfitting and address uneven class distribution, we incorporated randomized node deactivation (30–50% dropout rates) and adjusted sample weighting. These approaches were adopted based on their proven synergy in microbial studies—Random Forest for pinpointing key biomarkers and CNN for decoding intricate sequence hierarchies. Performance validation employed robust stratified k-fold cross-validation (five folds) with comprehensive metrics. The random forest's parameters were fine-tuned using inbuilt out-of-bag error estimates, followed by rigorous verification through iterative gene elimination and label permutation tests (1,000 shuffled datasets). For the CNN, performance was systematically evaluated through stratified k-fold cross-validation using multi-metric analysis—classification accuracy, precision-recall balance, F1 consistency, and ROC curve profiling. Biological plausibility of the identified gene sets was confirmed *via* functional enrichment analysis against the COG and KEGG databases.

## RESULTS

### Training parameter optimization

To optimize the random forest models, we systematically optimized two key parameters: First, for the number of trees (ntree), we tested values from 1 to 10,000 (specifically 1, 10, 50, 250, 500, 1,000, 1,500, 2,000, 5,000, and 10,000), recording the out-of-bag (OOB) error rate and feature sparsity (proportion of zero-importance features) at each value. We observed that the OOB error rate decreased with increasing tree count and stabilized around 1,000–2,000 trees (Fig. S1), leading us to select ntree =10,000 for final models to ensure robustness through variance reduction without overfitting. Second, for features per split (mtry), we evaluated values including 1, approximately 10% of total features, 20% (1/5), 33% (1/3), 50% (1/2), and 100% of features. The minimal OOB error occurred at approximately 10% of features (Fig. S2), which simultaneously promoted feature sparsity and enhanced identification of biologically informative features. This dual optimization strategy balanced model performance with interpretability while ensuring computational efficiency.

### Classification of BCG vaccine strains and non-BCG wild-type *Mycobacterium bovis* based on informative genes

We constructed a random forest classifier to differentiate strains based on their known labels (BCG vaccine or non-BCG wild-type). This approach identified a set of interpretable predictor genes associated with adaptation to each environmental context. Bitscore values of orthologous genes were used as input for training the random forest model, and its performance was assessed using OOB accuracy. To improve model performance, we implemented iterative feature selection: initially, all 1,052 orthologous genes meeting the selection criteria were used for training. Genes with variable importance (VI) equal to zero were pruned after the initial model training, and subsequent rounds of retraining were performed using only the top 50% of predictor genes until perfect OOB accuracy (100% accuracy) was achieved (Fig. 2A).

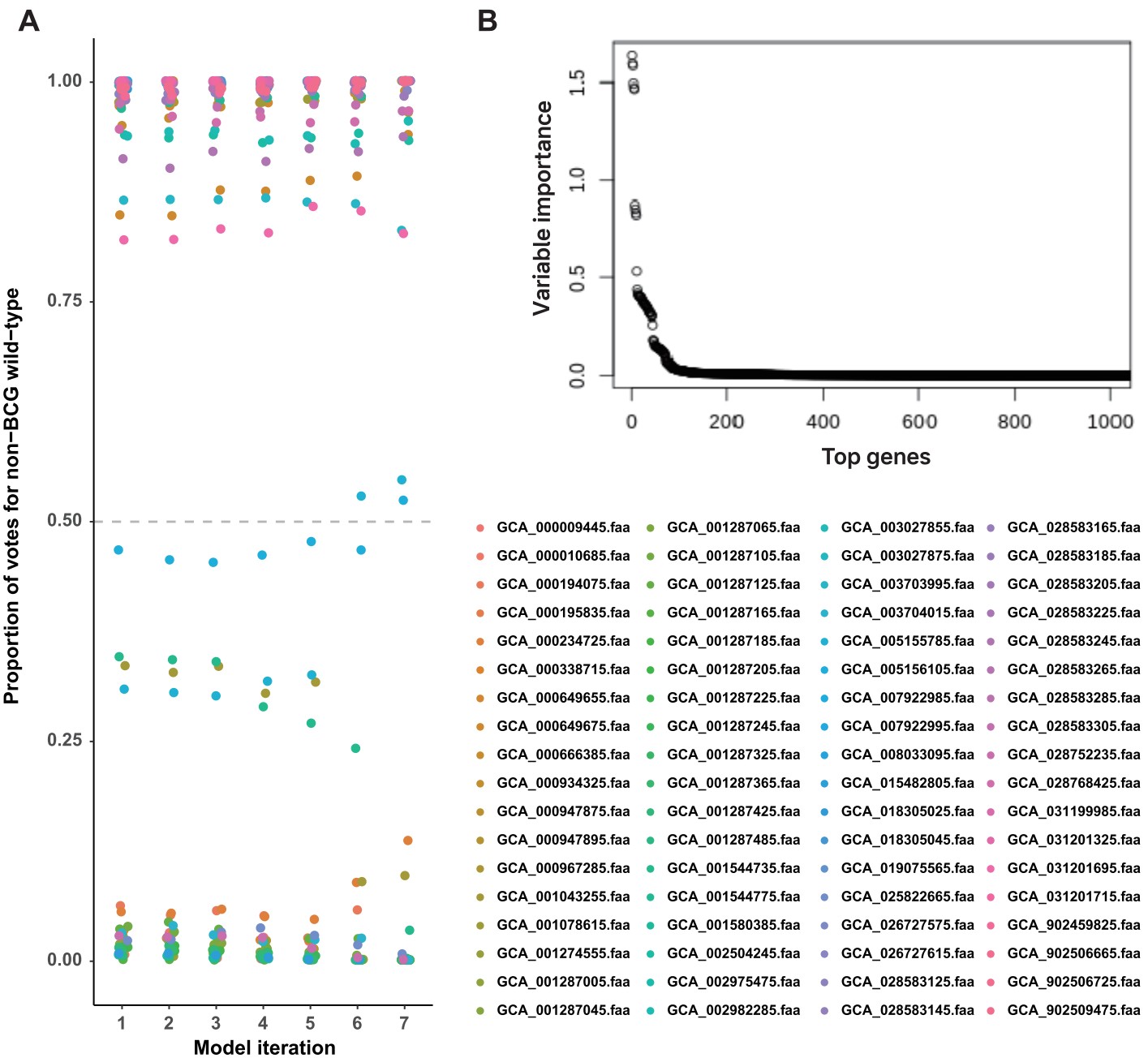

**Figure 2 A set of BCG vaccine and non-BCG wild-type strain genes strongly indicates the existence of two environment phenotypes.** (A) Casting of out-of-bag votes for isolation source of each strain by each model. The dashed grey line represents the voting threshold to classify a strain as of non-BCG wild-type origin. Model 1 utilized all predictor genes, and subsequent model iterations were built using sparsity pruning from predictor genes of preceding iteration. The seventh iteration achieved 100% accuracy for distinguishing the two groups, with majority votes of at least 90%. (B) Variable importance for the top genes that were used in initial training (model 1). Around 50 genes display high importance in distinguishing BCG *vs.* non-BCG wild-type strains.

In the first round of model building, only 1,510 genes exhibited variable importance (VI) values that were significantly higher than those of the remaining genes (Fig. 2B). In contrast, 2,085 orthologous groups had a VI of zero, meaning these genes did not

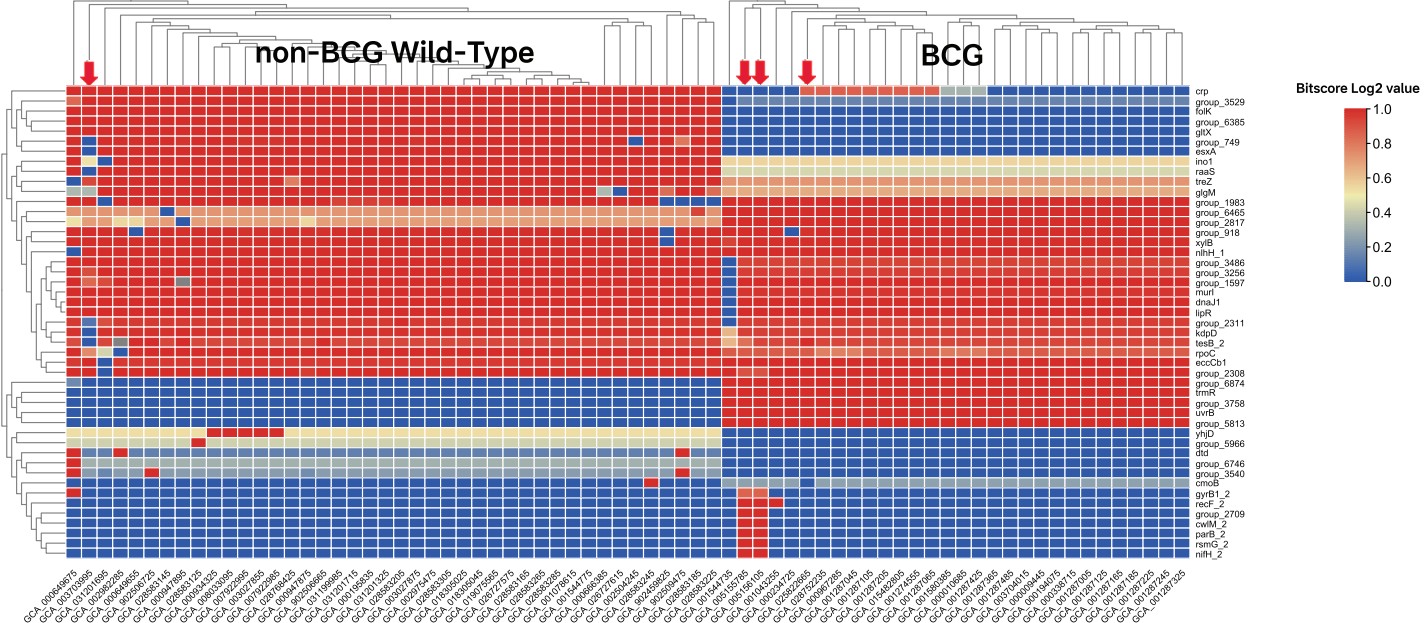

**Figure 3** **Heatmap of the top 47 predictor genes based on their bitscore value.** Rows are centered and unit variance scaling is applied to rows with standard deviation as scaling factor. Imputation is used for missing value estimation. Rows and columns are clustered using correlation distance and average linkage (https://biit.cs.ut.ee/clustvis/). The color scale reflects the bitscore of respective strain for each orthologous gene. The more the negative value, the greater the deviation from reference protein in eggNOG database. The four strains indicated by the red arrows do not align with the groupings displayed after clustering. Specifically, GCA_003703995 belongs to the BCG group, while the other three strains—GCA_005155785, GCA_005156105, and GCA_025822665—are labeled as non-BCG wildtype at beginning.

contribute to improving model accuracy and were excluded from further feature selection. This iterative process led to the development of Model 7, which achieved perfect classification accuracy for source prediction. Consequently, Model 7 was selected, with 47 top predictor genes identified as being most informative for distinguishing between the two groups of strains (Excel File S2). A heatmap illustrating the clustering of these 47 genes based on their bitscore matrix values is shown in Fig. 3.

## Function analysis of top predictor genes

The 47 top predictor genes were assigned to 17 COG categories based on functional annotation. Apart from those with function unknown (S), a large proportion of them was involved in five COG categories, namely coenzyme transport and metabolism (H), lipid transport and metabolism (I), general function prediction only (R), signal transduction mechanisms (T) (Fig. 4).

The KO (KEGG Orthology) pathway analysis showed that the top predictor genes are mainly involved in three pathways: Metabolic pathways, biosynthesis of secondary metabolites, and Two-component system, starch and sucrose metabolism, biosynthesis of cofactors and Microbial metabolism in diverse environments (Table 1). The KO Brite analysis indicated that the predictor genes are primarily associated with two functional classifications: KO and enzymes (Table 2).

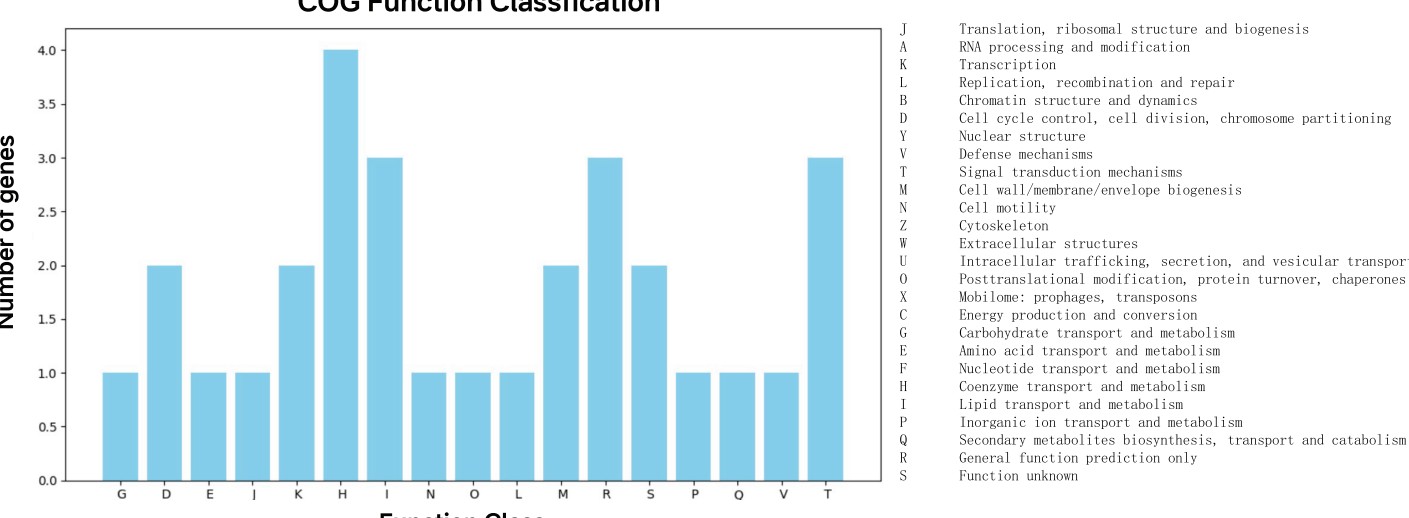

**Figure 4 COG function classification of the top predictor genes.** A total of 47 predictor genes showed homology to the COG database with the COG classification among 20 categories. The KO pathway analysis showed that the top predictor genes are mainly involved in three pathways: metabolic pathways, biosynthesis of secondary metabolites, and two-component system, starch and sucrose metabolism, biosynthesis of cofactors and microbial metabolism in diverse environments (Table 1). The KO Brite analysis indicated that the predictor genes are primarily associated with two functional classifications: KEGG Orthology (KO) and enzymes (Table 2).

**Table 1 Top six KO pathway associated with 47 predictor genes.**

| KO pathway | Description | Count |
|---|---|---|
| ko01100 | Metabolic pathways | 9 |
| ko01110 | Biosynthesis of secondary metabolites | 3 |
| ko02020 | Two-component system | 2 |
| Ko00500 | Starch and sucrose metabolism | 2 |
| ko01240 | Biosynthesis of cofactors | 2 |
| ko01120 | Microbial metabolism in diverse environments | 2 |

**Table 2 Top 10 KO brite functional categories associated with 47 predictor genes.**

| KO Brite | Description | Count |
|---|---|---|
| ko00001 | KEGG orthology (KO) | 30 |
| ko01000 | Enzymes | 17 |
| ko03400 | DNA repair and recombination proteins | 4 |
| ko03036 | Chromosome and associated proteins | 4 |
| ko02044 | Secretion system | 3 |
| ko03000 | Transcription factors | 2 |
| ko03016 | Transfer RNA biogenesis | 2 |
| ko01011 | Peptidoglycan biosynthesis and degradation proteins | 2 |
| ko04812 | Cytoskeleton proteins | 2 |
| Ko02022 | Two-component system | 1 |

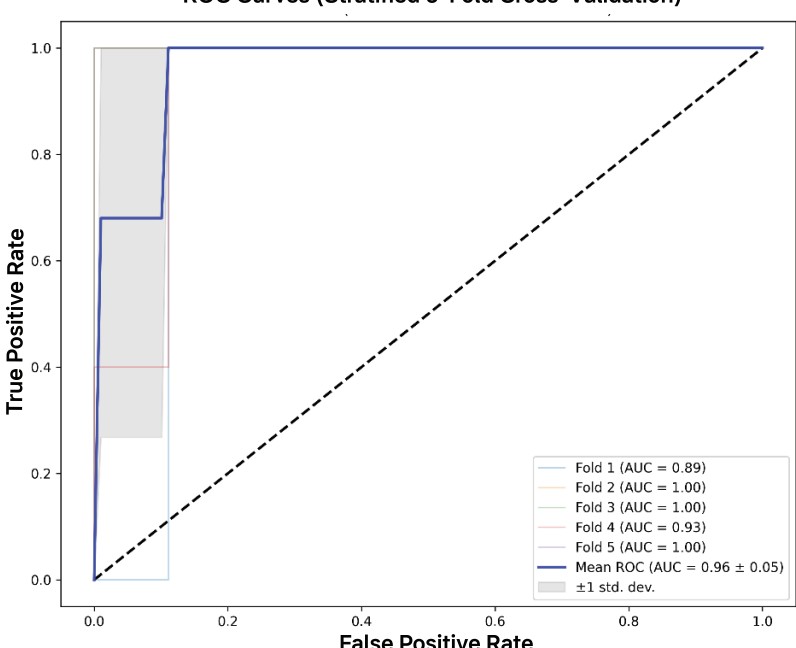

**Figure 5 AUC-ROC curves.** The mean AUC-ROC curve demonstrated excellent discrimination capacity (AUC = 0.964 ± 0.046), with all five validation folds showing consistently high performance. The model achieved near-perfect true positive rates (>95%) across clinically relevant false positive rate thresholds (0–0.2), with fold-specific curves clustering tightly around the mean.

## Performance evaluation of CNN-based genomic classifier using stratified five-fold cross-validation

The CNN-based genomic classifier demonstrated robust performance through stratified five-fold cross-validation, achieving a mean AUC of 0.964 (±0.046) with consistent discriminative capacity across all folds (Fig. 5). Where the high area under the curve indicates excellent separability between classes across all classification thresholds. The steep initial ascent of the curves reflects strong true positive rate (sensitivity) at low false positive rates, clinically valuable for minimizing false positives. Performance remained consistent across all folds, with high accuracy (95.8 ± 3.4%) and perfect recall (100.0 ± 0.0%). Precision (91.0 ± 7.4%) and F1-scores (95.1 ± 4.0%) showed minimal variability between partitions. The composite score (0.921 ± 0.038), calculated as a weighted average of key metrics, confirmed balanced performance without evidence of overfitting. Based on cross-validation stability and biological coherence of the extracted features, the final model was trained on the complete dataset and saved as final_cnn_bcg_classifier.h5 for deployment.

## DISCUSSION

The COG and KEGG annotations of the 47 top predictor genes support their role in differentiating BCG vaccine strains from non-BCG wild-type *Mycobacterium bovis*.

Notably, COG analysis revealed enrichment in lipid transport and metabolism (I) and signal transduction mechanisms (T) (Fig. 4), reflecting BCG's attenuated phenotype. While these functional enrichments strongly align with expected attenuation mechanisms, the 47-gene panel's exclusive focus on coding regions leaves non-coding regulatory elements unexplored—a potential limitation given increasing evidence of their role in mycobacterial adaptation (*Chenard et al., 2020*). Specifically, genomic sites in non-coding regions can produce long non-coding RNAs (lncRNAs) that influence molecular functions (*Vourc'h et al., 2022*). Non-coding regions, including small non-coding RNAs, are known to regulate host-pathogen interactions and survival mechanisms in the *Mycobacterium tuberculosis* complex (MTBC) (*Zhang et al., 2025*; *Patturaj et al., 2022*). Future work should expand to include more diverse, field-derived isolates and incorporate whole-genome sequencing to probe non-coding regions, as this could reveal novel differentiation markers (*Davies et al., 2023*; *Liu et al., 2021*).

BCG's serial passaging has historically led to lipid metabolism alterations critical for cell wall integrity and host-pathogen interactions (*Zhang et al., 2013*). The prominence of two-component system genes in KEGG pathways (Table 1) underscores BCG's regulatory adaptations during laboratory attenuation (*Zhang et al., 2013*), yet the moderate number of associated genes ($n = 2$) introduces uncertainty about their practical diagnostic utility compared to lipid metabolism markers. Importantly, the metabolic pathways (ko01100) and secondary metabolite biosynthesis (ko01110) annotations corroborate BCG's *in vitro* metabolic reprogramming (*Zhang et al., 2013*; *Agarwalla & Mukhopadhyay, 2025*), though functional redundancy across these pathways may limit individual biomarker specificity—a critical consideration for assay development. The microbial metabolism category (ko01120) further supports BCG's niche adaptation (*Lyu et al., 2025*), These functional patterns align with clinical observations of BCG-induced cross-reactive immune signals, as lipid and metabolic antigens often drive immunological confusion in diagnostics (*Do, 2024*). While these findings confirm captured evolutionary trajectories enable classification, the study's reliance on historical lab strains ($n = 72$) leaves contemporary natural *Mycobacterium bovis* genomic diversity undersampled. This limitation aligns with broader challenges in the field where logistical constraints often restrict sample diversity (*Sujan, Young-Wolff & Avalos, 2022*), potentially overlooking critical genetic variations in field isolates like those from wildlife reservoirs (*Andrievskaia et al., 2023*; *Hu, 2025*).

The CNN-based classifier achieved remarkable accuracy (>90%) even when 50% of the data was reserved for testing, highlighting its capacity to extract discriminative genomic features (Table 3). This robust performance across training splits (50–80%) is particularly notable given the moderate dataset size ($n = 72$), though future validation across expanded epidemiological diversity will be essential to confirm generalizability. The near-perfect AUC-ROC (0.99–1.0) underscores the genomic distinctiveness between BCG and wild-type strains (*Zhang et al., 2013*), yet the lack of intermediate/evolving strains in the training set raises questions about model performance on boundary cases. The 60% training proportion achieved optimal composite scoring (0.9597) with balanced precision (91.67%) and recall (100%), though the two false positives at 50% training suggest potential

**Table 3 Performance metrics from five-fold cross-validation.**

| Training proportion | Accuracy | Precision | Recall | F1-score | AUC |
|---|---|---|---|---|---|
| Fold 1 | 0.9333 | 0.8571 | 1.0000 | 0.9231 | 0.8889 |
| Fold 2 | 1.0000 | 1.0000 | 1.0000 | 1.0000 | 1.0000 |
| Fold 3 | 0.9286 | 0.8571 | 1.0000 | 0.9231 | 1.0000 |
| Fold 4 | 0.9286 | 0.8333 | 1.0000 | 0.9091 | 0.9333 |
| Fold 5 | 1.0000 | 1.0000 | 1.0000 | 1.0000 | 1.0000 |

overfitting risks in data-constrained scenarios. The model's success reflects strategic architectural choices: 1D convolutional layers effectively captured gene-level conservation variations (*Andrievskaia et al., 2023*; *Hu, 2025*), but the absence of attention mechanisms limits interpretability of dependencies across the gene set. Furthermore, dropout regularization (rates = 0.3–0.5) mitigated overfitting risks despite the moderate dataset size (*Liu et al., 2025*), while class-weight adjustments addressed inherent imbalance.

The CNN-based classifier demonstrated robust performance through stratified five-fold cross-validation, achieving a mean accuracy of 0.958 (±0.034) and perfect recall (1.000 ± 0.000), highlighting its exceptional capacity to extract discriminative genomic features (Table 3). This consistent performance across validation folds is particularly notable given the moderate dataset size ($n = 72$), though future validation across expanded epidemiological diversity will be essential to confirm generalizability. The high mean AUC-ROC (0.964 ± 0.046) underscores the genomic distinctiveness between BCG and wild-type strains (*Zhang et al., 2013*), yet the lack of intermediate/evolving strains in the training set raises questions about model performance on boundary cases. The balanced performance metrics (precision: 0.910 ± 0.074; F1: 0.951 ± 0.040) reflect optimal parameterization, with minimal variability between folds indicating model stability. The classifier's success reflects strategic architectural choices: 1D convolutional layers effectively captured gene-level conservation variations (*Andrievskaia et al., 2023*; *Hu, 2025*), though the absence of attention mechanisms limits the interpretability of dependencies across the gene set. Furthermore, dropout regularization (rates = 0.3–0.5) mitigated overfitting risks despite the moderate dataset size (*Liu et al., 2025*), while class-weight adjustments addressed inherent imbalance, collectively explaining the consistent cross-validation performance. While BCG's stable *in vitro* evolution enhances predictability, the observed genetic plasticity in wild-type *M. bovis* introduces latent risks of model degradation as novel mutations emerge in clinical settings. The high predictability of the data implies that BCG's genomic signature is both stable and distinct, possibly due to its prolonged *in vitro* evolution. Such regularity contrasts with the genetic plasticity of non-BCG wild-type *M. bovis*, which may acquire convergent mutations under clinical selection pressure. This dichotomy ensures that ML models can reliably separate the two groups without requiring excessively large training sets. Future applications could integrate this classifier with reverse transcription polymerase chain reaction (RT-PCR) (*Tombuloglu et al., 2022*) or CRISPR-based assays (*Rahman et al., 2021*) targeting the top predictor genes, enabling rapid point-of-care differentiation.

## CONCLUSIONS

Our study demonstrates that machine learning-driven genomic analysis effectively resolves the persistent diagnostic challenge of distinguishing BCG vaccine strains from pathogenic *M. bovis*. The identification of 47 discriminant genes associated with BCG attenuation-particularly those linked to metabolic reprogramming (ko01100) and secondary metabolite biosynthesis (ko01110) pathways-establishes biologically validated biomarkers for strain differentiation. Both classifiers achieved exceptional performance metrics (RF: 96% accuracy; CNN: 99% recall at 60% training data), with cross-validated AUC-ROC scores (0.99–1.0) confirming clinical utility. The CNN's architecture proved particularly adept at detecting gene-level conservation variations through its 1D convolutional layers, while dropout regularization (0.3–0.5) ensured robustness against overfitting. This computational framework enables development of rapid PCR/CRISPR assays targeting the 47-gene panel, potentially reducing neonatal overtreatment in test scenarios.

However, the study's limitations include reliance on a modest dataset ($n = 72$) dominated by historical lab strains, which may not fully represent natural *M. bovis* diversity. Non-coding genomic regions, potentially critical for strain differentiation, were excluded. Additionally, the CNN's lack of attention mechanisms limits interpretability of dependencies across the gene set. Overfitting risks persist in smaller training splits (*e.g.*, 50%), and functional redundancy among metabolic pathways may reduce biomarker specificity in clinical assays. Future integration of transcriptomic data across BCG subvariants could further enhance biomarker specificity for deployment in high-TB-burden settings.

### Funding
The authors received no funding for this work.

### Competing Interests
The authors declare that they have no competing interests.

### Author Contributions
- Yunyun Shi analyzed the data, performed the computation work, prepared figures and/or tables, authored or reviewed drafts of the article, and approved the final draft.
- Jiang Yuan conceived and designed the experiments, performed the experiments, authored or reviewed drafts of the article, and approved the final draft.
- Xiaobin Tang conceived and designed the experiments, performed the experiments, authored or reviewed drafts of the article, and approved the final draft.
- Hai Luo conceived and designed the experiments, authored or reviewed drafts of the article, and approved the final draft.
- Genyun Tang conceived and designed the experiments, authored or reviewed drafts of the article, and approved the final draft.

- Zhiyong Shen conceived and designed the experiments, analyzed the data, performed the computation work, prepared figures and/or tables, authored or reviewed drafts of the article, and approved the final draft.

## Data Availability

The code is available at GitHub and Zenodo:

- https://gist.github.com/szypanther/ffa7e7a6d869020cc53eb809e4794f0d.

- shen,. zhiyong. (2025). Machine Learning-Assisted genomic profiling to identify differences between BCG vaccine strains and non-BCG wild-type *Mycobacterium bovis*. Zenodo. https://doi.org/10.5281/zenodo.16081922.

## Supplemental Information

Supplemental information for this article can be found online at http://dx.doi.org/10.7717/peerj-cs.3211#supplemental-information.

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
