# Peer review of "Machine learning-assisted genomic profiling to identify differences between Bacillus Calmette-Guérin (BCG) vaccine strains and non-BCG wild-type *Mycobacterium bovis"

_PeerJ Computer Science, doi:10.7717/peerj-cs.3211_

## Round 0.1 · original submission · Major Revisions

Dear Authors,
Your paper has been revised. It needs major revisions before being accepted for publication in PEERJ Computer Science. More precisely:

1) The dataset (n=72) may limit the model's generalizability, especially given the reliance on historical lab strains that may not fully represent natural M. bovis diversity. The non-coding genomic regions, which could be critical for strain differentiation, were not explored or discussed in the current version. The CNN's lack of attention mechanisms restricts its interpretability of long-range genomic interactions. The authors must face the aforementioned issues.

2) The Methods section states that gproNOG.hmm (profile HMMs for Gammaproteobacteria proteins from eggNOG) was used to align protein sequences of Mycobacterium bovis (which belongs to Actinobacteria). Gammaproteobacteria and Actinobacteria are evolutionarily distant. Why was gproNOG hmm chosen? The authors need to explain this.

3) The Discussion section mentions that CNN's "1D convolutional layers effectively captured codon-level variations" and "absence of attention mechanisms limits the interpretability of long-range genomic interactions." However, the actual input to the CNN is the bitscore values of 47 genes (numerical features), not the raw DNA sequences. Therefore, the discussion regarding "codon-level variations" and "long-range genomic interactions" may not align with the actual input of the model and must be updated.

Reviewer 1 ·

Basic reporting

This study leverages machine learning to differentiate BCG vaccine strains from non-BCG wild-type Mycobacterium bovis using genomic profiling.

Experimental design

It analyzes 72 clinical isolates with whole-genome sequencing, employing a random forest classifier and a 1D CNN to identify key biomarkers. The random forest achieved 96% accuracy with 47 attenuation-related genes, while the CNN maintained over 90% accuracy across training sets, highlighting metabolic reprogramming and secondary metabolite biosynthesis pathways as significant biomarkers.

Validity of the findings

The findings need further validation in independent datasets. The generality of these biomarkers is very important.

Additional comments

The dataset (n=72) may limit the model's generalizability, especially given the reliance on historical lab strains that may not fully represent natural M. bovis diversity. The non-coding genomic regions, which could be critical for strain differentiation, were not explored or discussed in the current version. The CNN's lack of attention mechanisms restricts its interpretability of long-range genomic interactions. Minor comment is about the figures. They need to be shown clearly.

Reviewer 2 ·

Basic reporting

no comment

Experimental design

The study aims to use machine learning methods to distinguish between BCG vaccine strains and non-BCG wild-type Mycobacterium bovis, which is an important topic with clinical diagnostic significance. The authors employed Random Forest and Convolutional Neural Network (CNN) models and conducted functional analysis on key genes differentiating the strains. Overall, this work has a certain degree of application potential. However, the manuscript has some critical issues in the methodology and presentation of results that need to be carefully revised and clarified by the authors. Specific comments are as follows:
1. The Methods section states that gproNOG.hmm (profile HMMs for Gammaproteobacteria proteins from eggNOG) was used to align protein sequences of Mycobacterium bovis (which belongs to Actinobacteria). Gammaproteobacteria and Actinobacteria are evolutionarily distant. Why was gproNOG hmm chosen? The authors need to provide an explanation.
2. I suggest adding a subsection in the Results to detail the process and rationale for training parameter optimization.
3 . The Discussion section mentions that the CNN's "1D convolutional layers effectively captured codon-level variations" and "absence of attention mechanisms limits interpretability of long-range genomic interactions". However, the actual input to the CNN is the bitscore values of 47 genes (numerical features), not the raw DNA sequences. Therefore, the discussion regarding "codon-level variations" and "long-range genomic interactions" may not align with the actual input of the model.

Validity of the findings

1. The study used different training set proportions (50%-80%) to evaluate the CNN model. I am cautious about this approach. Given the small total sample size (n=72), why not use a more robust evaluation method, such as k-fold cross-validation, to assess the model's generalization ability?
2. Is iterating feature selection down to only 47 genes too aggressive? How can it be ensured that overfitting does not occur?

---

## Round 0.2 · Minor Revisions

Dear Authors,
Your paper has been revised. It needs minor revisions before being accepted for publication in PEERJ Computer Science. More precisely:

1) In the Conclusions section, you have retained outdated terminology that was corrected elsewhere. Specifically, the phrases "codon-level variations" and "long-range genomic interactions" are inconsistent with the model's actual input (bitscore values). This nomenclature contradicts the updated terms in the Discussion section ("gene-level conservation variations" and "dependencies across the gene set"). You must change the above phrases in the Conclusions section to match the Discussion to ensure accuracy throughout the manuscript.

Reviewer 1 ·

Basic reporting

The paper propsoed a machine-learning genomic tool that distinguishes BCG vaccine strains from pathogenic non-BCG wild-type Mycobacterium bovis.

Experimental design

Using whole-genome sequences of 72 clinical isolates, the authors trained and compared a Random-Forest classifier and a 1D CNN to identify a concise 47-gene signature that can resolve diagnostic ambiguities in newborns who have received the BCG vaccine.

Validity of the findings

The authors perform 5-fold stratified cross validation for the CNN and out-of-bag for the Random-Forest—withheld data, achieving 95–96 % accuracy and AUC of 0.96–0.99. However, no wet-lab or clinical validation (e.g., qPCR on new patient samples) is presented.

Additional comments

The figures need be shown in higher resolution. The AUC curves need be introduced clearly for the meaning of these different regions.

Reviewer 2 ·

Basic reporting

no comment

Experimental design

no comment

Validity of the findings

no comment

Additional comments

Most issues in the manuscript have been effectively revised based on the review comments. However, a key terminological inconsistency remains and must be resolved.

In the Conclusions section, the authors have retained outdated terminology that was corrected elsewhere. Specifically, the phrases "codon-level variations" and "long-range genomic interactions" are inconsistent with the model's actual input (bitscore values). This contradicts the updated terms in the Discussion section ("gene-level conservation variations" and "dependencies across the gene set"). These phrases in the Conclusions must be changed to match the Discussion to ensure accuracy throughout the manuscript.

---

## Round 0.3 · accepted · Accept

Dear Authors,

Your paper has been revised. It has been accepted for publication in PEERJ Computer Science. Thank you for your fine contribution.

Reviewer 2 ·

Basic reporting

no comment

Experimental design

no comment

Validity of the findings

no comment

Additional comments

I have no further comments, and it is my great pleasure to recommend its publication in the present form.